# Assessing the Activity under Different Physico-Chemical Conditions, Digestibility, and Innocuity of a GAPDH-Related Fish Antimicrobial Peptide and Analogs Thereof

**DOI:** 10.3390/antibiotics12091410

**Published:** 2023-09-06

**Authors:** Samuel Cashman-Kadri, Patrick Lagüe, Ismail Fliss, Lucie Beaulieu

**Affiliations:** 1Institute of Nutrition and Functional Foods (INAF), Université Laval, Québec, QC G1V 0A6, Canada; samuel.cashman-kadri.1@ulaval.ca (S.C.-K.); ismail.fliss@fsaa.ulaval.ca (I.F.); 2Department of Food Science, Faculty of Agricultural and Food Sciences, Université Laval, Québec, QC G1V 0A6, Canada; 3Québec-Océan, Université Laval, Québec, QC G1V 0A6, Canada; 4Department of Biochemistry, Microbiology and Bioinformatics, Université Laval, Quebec, QC G1V 0A6, Canada; patrick.lague@bcm.ulaval.ca; 5Institute for Integrative Systems Biology, Department of Biochemistry, Microbiology and Bio-Informatics, Pavillon, Alexandre-Vachon, Université Laval, 1045 Avenue de la Medecine, Québec, QC G1V 0A6, Canada; 6The Quebec Network for Research on Protein Function, Engineering, and Applications (PROTEO), 1045 Avenue de la Medecine, Québec, QC G1V 0A6, Canada

**Keywords:** antimicrobial peptides, AMPs, GAPDH, hemolytic activity, membrane permeabilization, SJGAP, antimicrobial activity, digestibility, cations, pH-dependence, salt resistance

## Abstract

The antimicrobial activity of SJGAP (skipjack tuna GAPDH-related antimicrobial peptide) and four chemical analogs thereof was determined under different physicochemical conditions, including different pH values, the presence of monovalent and divalent cations, and after a heating treatment. The toxicity of these five peptides was also studied with hemolytic activity assays, while their stability under human gastrointestinal conditions was evaluated using a dynamic in vitro digestion model and chromatographic and mass spectrometric analyses. The antibacterial activity of all analogs was found to be inhibited by the presence of divalent cations, while monovalent cations had a much less pronounced impact, even promoting the activity of the native SJGAP. The peptides were also more active at acidic pH values, but they did not all show the same stability following a heat treatment. SJGAP and its analogs did not show significant hemolytic activity (except for one of the analogs at a concentration equivalent to 64 times that of its minimum inhibitory concentration), and the two analogs whose digestibility was studied degraded very rapidly once they entered the stomach compartment of the digestion model. This study highlights for the first time the characteristics of antimicrobial peptides from Scombridae or homologous to GAPDH that are directly related to their potential clinical or food applications.

## 1. Introduction

For several years now, the issue of antibiotic resistance has been one of the major concerns of health authorities. Indeed, while multi-resistant bacteria are causing more and more infections, the discovery of new antibiotic molecules has significantly slowed down for several decades [1,2]. Thus, because of these two issues, it is estimated that by 2050, infections caused by antibiotic-resistant microorganisms will cause 10 million deaths annually if new approaches to fighting these microorganisms are not put forward [3,4]. For many years, there has been a huge effort from the scientific community to develop new antimicrobial agents that are effective, safe, and less susceptible to the phenomenon of antimicrobial resistance.

Antimicrobial peptides (AMPs) are among the molecules presenting an interesting potential as an alternative to classical antibiotics. Indeed, there are several databases specifically dedicated to AMPs in which more than 3000 peptides have been indexed, and a growing number of articles concerning these molecules are published every year [1,4]. AMPs are very diverse molecules made up of amino acids (normally between 10 and 50) and are part of the front lines of the immune defense system of a wide range of organisms, from the most primitive plants and invertebrates to human beings, and have inhibitory activity against microorganisms such as bacteria, fungi, parasites, and even viruses [2,5]. Most of these peptides share certain physicochemical properties, including moderate hydrophobicity and a net positive charge (at a neutral pH), but many neutral and anionic antimicrobial peptides have been discovered and studied [6,7].

There are many reasons for this strong interest in AMPs, including the fact that they have several advantages over traditional antibiotic molecules. Indeed, most antimicrobial peptides have non-specific and multi-hit mechanisms of action, targeting structures shared by several microorganisms, notably cell membranes [7,8,9]. This allows them not only to have a broad spectrum of activity and limit the development of microbial resistance but also to be effective against multi-resistant microorganisms against which conventional antibiotics have lost their effectiveness [5,7].

In addition, many antimicrobial peptides are highly effective in inhibiting Gram-negative bacteria, whereas the outer membrane of Gram-negative bacteria is often impermeable to traditional antibiotics, making them ineffective [4]. On another note, AMPs are also being studied for their promising potential as preservatives in food and even for the control of microbial pathogens in aquaculture and agriculture, which makes them versatile molecules [10,11].

Obviously, research on antimicrobial peptides for the development of therapeutic molecules or for other applications is not without its difficulties. Indeed, among the most frequent challenges associated with natural AMPs, there can be a loss of activity in the physiological environment or in food matrices (when peptides are to be used as food preservatives), notably because of the presence of mineral salts, a more or less high toxicity for mammalian cells, and being susceptible to degradation by various proteases present in the gastro-intestinal tract [1,3,4].

However, susceptibility to degradation by digestive enzymes may also be desirable, depending on the intended application of the AMPs. It is therefore very important to document the stability of AMPs under different physicochemical conditions and their toxicity in order to develop molecules of pharmaceutical or food interest. The possibility of chemically modifying natural AMPs to improve their therapeutic index, which is defined as the ratio of the antimicrobial activity to the hemolytic activity [12], stability, or to synthesize de novo peptides, is also increasingly explored [2,5,6].

Fish and other aquatic organisms are very good sources of natural AMPs, as shown by several reviews of the scientific literature that have been published on this subject [13,14,15,16]. In fact, fish are highly exposed to pathogenic microorganisms in their environment, and AMPs are an important component of their immune defense system to fight against these pathogens. Several studies have investigated the antimicrobial activity of some of these AMPs from fish under different conditions, their hemolytic activity, and their digestibility.

For example, pleurocidin and analogs thereof have been shown to have little or no hemolytic activity [17]. Two AMPs from Atlantic cod, named Gad-1 and Gad-2, showed weak hemolytic activity, but the antibacterial activity of Gad-1, unlike that of Gad-2, was shown to be sensitive to pH changes [18]. Finally, it was shown that two AMPs belonging to the large family of piscidins, namely moronecidin and myxinidin, retain their antimicrobial activity in the presence of different salts, depending on the concentrations tested [19,20].

Recently, research studies have been conducted on the natural fish AMP named “skipjack tuna glyceraldehyde-3-phosphate dehydrogenase (GAPDH)-related peptide (SJGAP)” [21], including a structure-activity and mechanistic study that was able to highlight the antibacterial and antifungal activity of this peptide and chemical analogs and their ability to interact with and permeabilize microbial membranes [22]. This study established links between the chemical structure of SJGAP and analogs thereof, their antimicrobial activity, and their mechanism of action, which had not yet been done for the GAPDH homologous AMPs, although GAPDH-related AMPs have been studied in recent years [23,24,25,26,27,28]. For example, it has been shown that the antimicrobial action of the SJGAP and its analogs is intimately related to their cationicity and hydrophobicity, and these peptides induce membrane permeabilization both in bacterial and fungal strains [22].

The hemolytic activity of SJGAP and its antimicrobial activity in the presence of NaCl, proteases, or at different pHs have been slightly documented. However, they have not been studied in relation to their chemical structure with the use of chemical analogs [21]. It is therefore very important to further investigate these points to establish relationships between the chemical structure of these peptides, their stability and potency under different physicochemical conditions, and their sensitivity to digestive enzymes in a dynamic context. This will help to better evaluate their potential applications, whether as therapeutic molecules or as preservatives to inhibit the growth of microbial pathogens and spoilage agents in foods.

The aim of this study is to further document the potential application of SJGAP and chemical analogs thereof as therapeutic molecules or food antimicrobial agents. To do this, several objectives were pursued. First, (i) the impact of different physicochemical conditions (presence of monovalent and divalent cations, pH, and heating treatment) on the antimicrobial activity of SJGAP and its analogs was determined, and (ii) their toxicity was assessed by measuring their hemolytic activity. This allowed us to (iii) establish relationships between the chemical structure of these analogs and their stability under different physicochemical conditions. Finally, (iv) the susceptibility of the peptides to digestive enzymes was evaluated in a dynamic model of the human gastro-intestinal tract, providing information on their safety in relation to their possible accumulation and potential routes of administration.

## 2. Results

### 2.1. Antimicrobial Activity

Minimal inhibitory concentrations (MICs) and minimal bactericidal concentrations (MBCs) of peptide analogs against *E. coli* ATCC 11229 in the presence of 1, 5, and 10 mM CaCl_2_ and MgCl_2_ are shown in Table 1, while the results obtained with 50, 150, and 300 mM of NaCl and KCl are presented in Table 2. Table 3 presents the MICs and MBCs of the peptides against the same bacteria at different pH values and after a heating treatment (100 °C, 15 min).

From Table 1, it can be seen that all peptide analogs are very sensitive to the presence of divalent cations; at a concentration of 1 mM of CaCl_2_ or MgCl_2_, they completely lose their activity. The results presented in Table 2 show that peptides are also sensitive to monovalent cations (Na^+^ and K^+^), but to a lesser extent. Specifically, peptide analogs 5 and 6 remain active in the presence of 50 mM NaCl or KCl but lose their activity when this concentration reaches 150 mM. Analogs 1 and 7 even seem to benefit from the presence of monovalent cations. Indeed, analog 1 is inactive in MHB (negative control), but shows antibacterial activity at a concentration of 128 μg/mL when in the presence of 150 and 300 mM of NaCl. Analog 7 showed no activity in non-modified MHB, but at concentrations of 64 μg/mL and 128 μg/mL, it showed inhibitory activity with the addition of 50 and 150 mM of KCl or NaCl, respectively. However, these concentrations cannot be rigorously identified as MICs, as very slight bacterial growth could be detected in the wells associated with these concentrations. Thus, there is evidence of a strong but not complete inhibition of bacterial growth. However, analog 8 was very salt-sensitive; it lost its activity upon addition of 50 mM NaCl or KCl to MHB.

Table 3 clearly shows that the peptide analogs are more strongly active at an acidic pH. Indeed, all the tested peptide analogs are strongly active at pH 5 (MICs of 2 to 8 μg/mL), which represent much lower values than those obtained at higher pHs, including in unmodified MHB. At pH 6, analogs 1, 5, and 7 are more active than at neutral (7) or basic (8) pH, but analog 6 showed an equivalent MIC value to those obtained at higher pHs and in the negative control. Analog 7 has produced some interesting results. Indeed, while it is totally inactive at neutral (including unmodified MHB) or basic pH values, it is strongly active at pH 5 and 6, totally inhibiting bacterial growth at a concentration of 8 μg/mL. However, this activity ceased at a concentration of 64 μg/mL, meaning that the inhibition produced by analog 7 was observed in wells in which it was concentrated between 8 and 32 μg/mL, inclusively. Another point to note in relation to the influence of pH on the activity of the peptide analogs is that a trend emerges with respect to their bactericidal activity: although showing lower MICs at acidic pH values, the bactericidal activity of analogs 1, 6, and 8 seems to disappear or decrease at these lower pH values compared to the results obtained in the tests performed at pH 7 and 8 (except for analog 8, whose bactericidal activity was also suppressed at pH 8) and in unmodified MHB. Regarding thermolytic resistance, all analogs were affected by a heating treatment of 100 °C for 15 min, but to different extents. Indeed, while analog 5 lost its antimicrobial activity following this treatment, analogs 6 and 8 retained inhibitory activity but showed twice as high values of MICs and MBCs compared to those obtained with the negative control.

### 2.2. Hemolytic Activity

The hemolytic activity of peptides 1, 5, 6, 7, and 8 at the different concentrations tested (0.5 to 1024 μg/mL) is shown in Figure 1. The hemolytic activity of the positive control (TritonX-100, 1%) was defined as 100%. Only peptide 5, at concentrations of 1024 and 512 μg/mL, showed hemolytic activities (14.1% and 11.8%, respectively) significantly higher than the values obtained with the negative control (PBS). The other four peptides, as well as peptide analog 5, at concentrations below 512 μg/mL, did not show any significant hemolytic activity compared to the negative control (PBS).

### 2.3. Digestibility of Peptide Analogs 6 and 8 in a Dynamic Gastro-Intestinal Model

The digestibility of peptide analogs 6 and 8 was evaluated in a dynamic model (TIM-1) that simulates human gastro-intestinal conditions. Overlays of fast protein liquid chromatography (FPLC) spectra of the initial meal and samples collected from the stomach compartment at times 0 and 20 for analogues 6 and 8, respectively, are shown in Figure 2 and Figure 3. Figure 2 clearly shows that peptide analog 6 is rapidly degraded by digestive enzymes. Indeed, the spectrum of the initial meal shows a well-defined peak at 17.6 mL, representing the undegraded native peptide, and a little peak at about 1.5 mL, representing by-products of a slight degradation of the peptide during sample preparation.

The spectrum of the sample from the stomach compartment at time 0 shows that this peak (17.6 mL) is transformed into a poorly defined shoulder centered at about 16 mL, with a much lower area under the curve value, showing strong degradation of the initial peptide. The spectrum of the sample from the stomach compartment after 20 min is similar but shows a decrease in the area under the curve of the shoulder centered at about 16 mL, indicating more severe degradation of the initial peptide. In these two spectra of the samples from the stomach compartment, one can also see the marked increase in the area under the curve of the peak initially present at 1.5 mL, indicating a greater number of molecules less retained by the column originating from hydrolysis of the initial peptide.

No traces of analog 6 could be detected in the FPLC spectra of samples from the duodenum compartment. MS/MS analyses confirm these results obtained by FPLC. In fact, the native peptide 6 sequence was not detected in the samples from the stomach compartment at T0 and T20. The longest sequence detected after 20 min of digestion in the stomach compartment (both replicates included) contains 22 amino acid residues (I_11_-K_32_), and its signal represents 0.00372% of all peptide signals detected in these samples (relative signal abundance).

Figure 3 shows a superposition of spectra quite similar to that in Figure 2, showing pronounced degradation of the peptide analog 8 in the gastro-intestinal tract. Indeed, in the spectrum of the initial meal, the well-defined peak at about 17 mL comes from the intact analog 8, while again, a much smaller peak can be seen at 1.5 mL from a slight degradation of the peptide prior to its passage through the digestion apparatus. A trace of this initial peak at 17 mL is still visible in the spectrum of the sample from the stomach compartment at T0, whereas it becomes almost invisible in the spectrum of the sample from the same compartment at T20. In both of these spectra, the appearance of a peak at 14 mL can also be seen, representing a peptide or group of peptides from the degradation of analog 8. Then, similar to what happened during the digestion of analog 6, the peak at 1.5 mL increases in magnitude during the digestion process.

No traces of analog 8 could be detected in the FPLC spectra of samples from the duodenum compartment. Similar to what was obtained for analog 6, MS/MS analyses confirm that analog 8 is highly sensitive to digestive enzymes, as its full sequence was not detected in samples from the stomach compartment at T0 and T20. The three longest sequences detected in the stomach compartment at T20 (both replicates included) contain 16 amino acid residues (L_14_-A_29_, V_1_-T_16_, and V_15_-I_30_) and together represent a relative signal abundance of 1.53%.

## 3. Discussion

Antimicrobial peptides represent one of the most promising and explored avenues for the discovery of new antibiotic agents. Their broad spectrum of action and their low propensity to stimulate the emergence of microbial resistance are certainly two of their most interesting advantages, not to mention that they also generally have a favorable safety profile [5]. Indeed, antimicrobial peptides generally have mechanisms of action that involve interaction with several molecular targets of varying affinity, unlike many classical antibiotic molecules that have a well-defined molecular target of high affinity, which is much less conducive to the emergence of resistant microorganisms [29]. In fact, AMPs act first and foremost on the cytoplasmic membrane components of microbial agents, notably phospholipids.

However, despite these significant advantages, there are several challenges when it comes to applying AMPs in clinical or industrial situations. Production costs related to peptide synthesis are obviously an issue to consider, but several studies have shown that certain antimicrobial peptides, which are very active under standard in vitro test conditions, become significantly less active when tested under more realistic application conditions, for example, in the presence of salts at physiological concentrations [5,29]. The pH can also have a marked influence on their activity, which can become important in relation to their possible applications as AMPs can be used in many different conditions [30]. For example, the skin represents an acidic environment, which can limit the possibilities of topical application for peptides showing low activity at low pH values [31]. Furthermore, some AMPs can also show hemolytic activity at the concentrations at which they have antimicrobial activity, which limits their possibility of use in a clinical context.

In addition, for some applications where AMPs are found in the gastro-intestinal (GI) tract, the resistance of AMPs to proteases, e.g., trypsin and chymotrypsin, may be a positive trait (if the locus of action of the peptides is in the GI tract) [3,4], but being susceptible to degradation by these enzymes may also be a good thing if they are to be used as food preservatives, as their activity is not desired in the gut [32,33]. It is therefore very important to know the susceptibility of AMPs to these proteases in terms of their potential applications. Finally, the resistance of AMPs to high temperatures is important to consider in relation to applications in the food industry since several foods must undergo heat treatments during their production [34,35].

### 3.1. Impact of pH, Salts, and Heat Treatment

Taking into account this context of the application of AMPs, the influence of monovalent and divalent cations on the antimicrobial activity of SJGAP and its analogs was tested. The concentrations of the salts used (NaCl, KCl, CaCl_2_, and MgCl_2_) were selected on the basis of recent work in the literature and the physiological concentrations of these cations, which are around 100–150 mM for sodium and potassium and 1–2 mM for calcium and magnesium [36,37,38].

Results obtained in this study show that the five tested peptide analogs are not affected in the same way by the presence of salts. In fact, results presented in Table 2 and Table 3 show that the presence of both monovalent and divalent cations negatively impacts the antimicrobial activity of these peptides, but the impact of divalent cations (Ca^2+^ and Mg^2+^) is much stronger than the effect of Na^+^ and K^+^. This is not very surprising, as it is well known that the presence of salts can significantly weaken the electrostatic interactions between AMPs and microbial membranes, which are central to their initial attachment and destabilizing action [4,5,39]. This could probably explain why analogs 5 and 6, and 8 to an even greater extent, lost their activity upon the addition of NaCl and KCl. In addition, other phenomena may also explain this impact of cations on the activity of AMPs, including modification of the three-dimensional structure of peptides (modification of intramolecular interactions) [6] and an impact on the organization of the polar heads of phospholipids on the surface of membranes [36]. The much greater impact of divalent cations on all the tested analogs can perhaps be explained by their direct competition with peptides for binding with the anionic components (phospholipids, lipopolysaccharides, etc.) of microbial membranes [40].

Regarding the interactions between cations and membrane phospholipids just mentioned, certain points can be developed to better understand the impact they may have on peptide analog activity. Na^+^ and Ca^2+^ ions are known to interact strongly with ester moieties of phospholipids, including phosphatidylethanolamines and phosphatidylglycerols (which are the main components of bacterial membranes [41]), whereas Mg^2+^ more often binds directly to the phosphate groups [42,43]. These ion-phospholipid interactions can alter their sensibility to AMPs in two main ways. Firstly, it has been shown that the binding of cations to phospholipids causes a decrease in the zeta potential of membranes, in that the overall negative charge of the membrane surface decreases as the cation concentration increases. This also masks the negative charge of the membrane surface due to the attraction of counter-anions to the cations adsorbed at the membrane interface [44]. Secondly, the adsorption of cations onto membranes is due to coordination mechanisms rather than electrostatic attraction, which has a direct impact on the hydrogen bonds between phospholipids and leads to significant compaction of membrane lipids and changes in the angle of their polar heads, thus affecting interactions with peptides [45,46]. K^+^ ions are also known to bind to membrane phospholipid head groups in the same way, but to a lesser extent than Na^+^, mainly due to their significantly larger size [47].

Many other AMPs listed in the literature have their antimicrobial activity partially or totally inhibited by various concentrations of different cations, including monovalent cations such as Na^+^ and K^+^ and divalent cations such as Ca^2+^ and Mg^2+^ [36,38,40]. Conversely, several AMPs have also shown a more or less high tolerance to the presence of salts, thus maintaining their antimicrobial activity in the presence of different cations [34,39,48]. It has also been observed that many of the antimicrobial peptides with good resistance to the presence of salts are of marine origin, thus opening the door to an evolutionary explanation for the emergence of these peptides [39]. The literature even presents cases of AMPs whose antimicrobial activity is increased in the presence of different salts [37], which is the case for analogs 1 and 7 tested in this study. Interestingly, these two analogs have the same net charge of +4, unlike analogs 5, 6, and 8, which have a stronger cationic character. Thus, it is possible that the interactions of these two analogues with microbial membranes are less related to electrostatic interactions, which could explain why cations have a less negative impact on their antimicrobial activity. It is even possible that the presence of salts has an influence on their secondary structure, allowing them to interact more efficiently with these membranes, for example, with hydrophobic interactions, as it is known that salt concentrations can have a direct impact on interactions between peptides and lipids [6]. This could explain why these peptides only show antibacterial activity when salts are added to the culture media.

The pH is also well known to have a significant impact on the antimicrobial activity of many peptides. Changes in pH can directly influence the electrostatic interactions between AMPs and microbial membranes. Indeed, changes in pH can alter the protonation states of different amino acid residues constituting the peptides, thus having a direct impact on their net charge. The ionizable components of microbial membranes can also have different charge states depending on the environmental pH value and have a significant impact on their interactions with ions, peptides, or proteins. For example, cardiolipins, which form up to 10% of the phospholipidome of *E. coli* [49], contain two ionizable phosphate groups that are normally both in dissociated form at physiological pH (the cardiolipin therefore takes the form of a dianion). This negative charge favors cation coordination and electrostatic interactions with cationic molecules at the interface, such as AMPs [50].

These factors can thus have a major influence on the initial interactions between AMPs and microbial membranes, which can drastically modify their activity [6]. More precisely, it is notably the histidine, aspartic acid, and glutamic acid residues that can be protonated under acidic conditions, which will increase the cationic character of the AMPs that contain them, thus promoting interactions with the anionic components of microbial membranes [31]. It can also be noted that alpha helices containing these residues can be stabilized at low pH values, which may also explain the impact on the antimicrobial activity of AMPs with this type of secondary structure [31]. In fact, several AMPs, including many particularly rich in histidine residues, have increased antimicrobial activity at acidic pH [30,51,52], although other AMPs less affected by pH changes are also documented in the literature [34].

These impacts of low pH values can most likely explain the fact that all analogs tested in this study are more active in acidic conditions. Indeed, all analogs have a histidine residue (position 21), which is located at the very end of the central alpha helix in peptides [22]. Therefore, it is plausible that both an increase in net charge due to protonation of this residue and the stabilization of the alpha helix explain an increase in their antimicrobial activity at pH 5 and 6. It is also very interesting to note that analogs 1 and 7, which did not show antibacterial activity in unmodified MHB, showed comparable MIC values to analogs 5 and 6 at pH 5 and pH 6. The fact that these two analogues are active at these acidic pHs can likely be explained by the presence of glutamic acid (position 26) and aspartic acid (position 32) residues, which have been substituted in analogs 5 and 6. It is also intriguing to note that analog 1, although inactive in unmodified MHB, showed antimicrobial activity when tested in MHB adjusted to pH 7 and 8. This may be due to the ionic strength provided by the buffers used to adjust the pH of the MHB, similarly to what was noted above for the activity of this same analog in the presence of 150 and 300 mM NaCl.

Regarding the loss of activity of analog 7 at a concentration of 64 μg/mL at pH 5 and 6, the hypothesis of peptide aggregation can be suggested. Indeed, given the lower net charge (+4) of this analog and its greater hydrophobicity, it is possible that at neutral and basic pH, intermolecular hydrophobic interactions are dominant between the peptide monomers, favoring their aggregation and thus limiting their antimicrobial activity [53]. Such a pH-dependent aggregation has already been observed for a peptide with immunomodulatory activity [54] and the LAH4 AMP [55]. However, given the protonation of some residues under acidic conditions, as explained above, the net charge of the peptides increases, promoting electrostatic repulsion between the monomers and interactions with anionic membrane components, thus reducing their aggregation and allowing their interactions with microbial membranes. It is possible that above a certain concentration (here, 64 μg/mL), intermolecular interactions between peptide monomers become dominant again, thus inducing their aggregation and the loss of antimicrobial activity. This aggregation phenomenon could also explain the loss of activity of analog 5 following the 15-min heat treatment at 100 °C. Indeed, as this analog is the one with the highest hydrophobicity, it is possible that an increase in temperature favored intermolecular hydrophobic interactions between the peptide monomers, causing their aggregation and the loss of antimicrobial activity.

### 3.2. Hemolytic Activity

The measurement of the hemolytic activity of membrane-active antimicrobial compounds is often used as an initial toxicity test, particularly due to the ease of isolating erythrocytes [56]. Of the peptides tested in the present work, only analog 5, which is the most hydrophobic of the five analogs included in this study (GRAVY = 0.603), showed significant hemolytic activity at the two highest tested concentrations (1024 and 512 μg/mL). These concentrations are much higher than the MIC of the same analog (16 μg/mL). These results are in full agreement with the trend often noted in the literature that the hemolytic activity of AMPs is strongly correlated with their hydrophobicity due to their hydrophobic interactions with erythrocyte membranes [48,56,57,58]. Indeed, unlike the negatively charged membranes of bacteria, mammalian cell membranes are weakly charged, being mainly composed of zwitterionic phospholipids and cholesterol [56].

### 3.3. Stability in Gastro-Intestinal Conditions

Resistance to the action of proteolytic enzymes is another characteristic that is important to study in relation to the potential application of AMPs. Indeed, from a clinical point of view, the susceptibility of AMPs to proteases can greatly limit their development, as it significantly reduces their half-life in human serum [2,3]. In addition, being sensitive to the action of enzymes present in the human gastro-intestinal tract can greatly limit their action in situ, which is why much research has been conducted to improve the bioavailability of antimicrobial peptides. For example, work on encapsulating nisin has been carried out in order to limit its degradation in the gastro-intestinal tract and combat the pathogenic bacterium *Clostridium difficile* in the colon [5,59]. On the other hand, if AMPs are used more as food preservatives, their degradation in the gastro-intestinal tract is desirable. In such a case, demonstrating that an AMP is degraded by proteolytic enzymes in the stomach or intestine is evidence of its safety, as it is not absorbed by the body and does not produce residue accumulation [11,60,61]. Analogs 6 and 8 were selected to test their digestibility. Indeed, analog 6 showed potent antimicrobial activity against different types of microbial strains (Gram− and Gram+ bacteria and fungi) [22] while not showing significant hemolytic activity (Figure 1). Analog 8, on the other hand, has the same primary sequence as native SJGAP but shows more interesting antimicrobial activity due to its C-terminal amidation.

The degradation of peptide analogs 6 and 8 in the stomach compartment was expected given the presence of pepsin in this compartment. Indeed, this enzyme is known to hydrolyze peptide bonds involving leucine, phenylalanine, and tyrosine [62], which are included three times in these peptides’ sequences (F_8_, L_14_, and F_20_). The fact that peptide degradation is already well advanced in the stomach compartment at T0 shows that these peptides are very sensitive to pepsin action. Indeed, this result means that after the withdrawal of the samples in this compartment at the very beginning of the digestion (T0), pepsin had enough time to act and degrade analogs 6 and 8 during the subsequent heating step aimed at inactivating the enzymatic activity since a delay of a few minutes was necessary before the sample reached 70 degrees.

The complete degradation of these peptides in the duodenum was also expected given the presence of pancreatin, which is a mixture of five digestive proteases: trypsin, chymotrypsin, elastase, carboxypeptidase A, and carboxypeptidase B [32]. Carboxypeptidases A and B are exopeptidases that break down peptide bonds located at the carboxyl end. Carboxypeptidase A cleaves bonds next to amino acids with aromatic, neutral, or acidic side chains, while carboxypeptidase B mainly cleaves bonds next to basic amino acids such as arginine and lysine. Analog 6, having a lysine residue at its C-terminal end, thus presents a cleavage site for carboxypeptidase B, while analog 8 presents a cleavage site for carboxypeptidase A because of its D_32_ residue, even though C-terminal amidation can increase resistance to exopeptidase degradation [63,64]. Elastase is an endopeptidase that specifically cleaves bonds near uncharged amino acids such as alanine, glycine, and serine, while chymotrypsin is another endopeptidase that hydrolyzes bonds adjacent to aromatic amino acids such as phenylalanine, tyrosine, and tryptophan. Finally, trypsin is another endopeptidase known to have a mechanism of action that targets peptide bonds on the carboxyl side of lysine and arginine residues [32,62]. Thus, these five enzymes contained in pancreatin are provided with multiple cleavage sites in the sequence of peptide analogs 6 and 8. The sensitivity of SJGAP (the native peptide on which the design of the peptide analogs conceptualized in the present study is based) to the action of trypsin and chymotrypsin had also been documented previously [21].

## 4. Materials and Methods

### 4.1. Peptide Design and Synthesis

SJGAP [21] was selected as the reference AMP for this study. Native SJGAP and four peptide analogs, which were designed as previously described [22], were synthesized by the external company Bio Basic (Markham, ON, Canada) with >95% purity, as confirmed by HPLC/MS reports. These synthetic peptides were used for the antimicrobial and hemolytic activity assays. For the in vitro digestibility assays, peptide analogs 6 and 8 were selected based on their physicochemical properties and antimicrobial and hemolytic activities. They were synthesized by the external company Innodal (Québec, QC, Canada) with >95% purity, as confirmed by HPLC/MS reports. The five peptide analog sequences, net charges, pI, molecular weights, and GRAVY indexes are presented in Table 4. Lyophilized peptide powders were stored at −20 °C until use.

### 4.2. Antimicrobial Activity Assays

#### 4.2.1. Bacterial Strain and Culture Conditions

The indicative bacterial strain *Escherichia coli* ATCC 11229 was selected to test the antimicrobial activity of SJGAP and analogs thereof under different physicochemical conditions, as it showed to be the most sensitive strain in a previous study [22]. The bacterial strain was stored in 25% (*v/v*) glycerol at −80 °C until use. It was cultivated in tryptic soy broth (TSB) (BD, Sparks, NV, USA) at 37 °C and subcultured three times before being used for antimicrobial activity assays.

#### 4.2.2. Minimal Inhibitory and Bactericidal Concentrations (MICs and MBCs)

MICs for *Escherichia coli* ATCC 11229 were determined using a broth microdilution protocol [65]. Briefly, twofold dilutions of peptide solutions (0.256 μg/mL) were made in Mueller Hinton Broth (MHB) (Oxoid, Basingstoke, UK) in 96-well polypropylene clear round bottom microplates (Corning Inc., Corning, NY, USA). Bacteria were cultured for 18 h and then diluted in MHB broth and inoculated in the microplates so that the final inoculum in the wells was 5 × 10^5^ CFU/mL. The final volumes in the wells were 200 μL. Plates were then incubated for 16 h at 37 °C. The MICs were considered the lowest peptide concentrations that totally inhibited visible bacterial growth in the wells. Chloramphenicol (Sigma-Aldrich, St. Louis, MI, USA) was used as a positive control to ensure the validity of the tests. After determining the MICs in each plate, 20 μL of the wells that showed no visible bacterial growth were inoculated on tryptic soy agar (TSA) plates, which were then incubated at 37 °C for 24 h. MBCs were defined as the lowest peptide concentrations that inhibited 99.9% or more of bacterial growth based on the initial 5 × 10^5^ CFU/mL inocula in the wells [66]. Three independent repetitions were completed, and two technical replicates were done for each independent repetition.

#### 4.2.3. Impact of Salts, pH, and Heating on Antimicrobial Activity

The impact of different salts on the antimicrobial activity of SJGAP and analogs thereof was tested by determining the effect on their MIC and MBC values. The same procedure as described above was used, but different concentrations of salts were added to the MHB medium in the microplates. To test the impact of monovalent cations, NaCl (Fisher Scientific, Fair Lawn, NJ, USA) and KCl (EMD, Gibbstown, NJ, USA) were tested at final concentrations of 50, 150, and 300 mM in the wells. Next, to assess the impact of divalent cations, CaCl_2_ (Fisher Scientific, Fair Lawn, NJ, USA) and MgCl_2_ (Fisher Scientific, Fair Lawn, NJ, USA) were tested at final concentrations of 1, 5, and 10 mM in the wells.

The antimicrobial activity of SJGAP and analogs thereof was tested at pH 5, 6, 7, and 8. For this purpose, the MHB medium used in the microplates for MIC and MBC testing was adjusted to pH 5 or 6 with 50 mM citrate buffer and pH 7 or 8 with 50 mM phosphate buffer.

The stability of the peptides at elevated temperatures was also tested. Mixtures of the peptides solubilized in MHB medium were heated to 100 °C for 15 min in a water bath. After cooling in an ice water bath, the solutions were loaded into the microplates and inoculated with the tested bacteria to conduct the MIC and MBC assays. For all tested conditions, three independent repetitions and two technical replicates for each of these repetitions were completed.

### 4.3. Hemolytic Activity Assays

The hemolytic activity of SJGAP and its analogs was determined spectrophotometrically by measuring the hemoglobin loss from horse erythrocytes based on procedures already described in the literature [40,67,68]. Briefly, fresh defibrinated horse blood (Quad Five, Ryegate, MT, USA) was centrifuged (2000× *g*, 5 min) and washed with phosphate-buffered saline (PBS) five times, then resuspended in PBS at 10% (*v/v*). Then, 50 μL of PBS were added to the wells 2 to 12 of 96-well polypropylene clear round bottom microplates (Corning Inc., Corning, NY, USA). A volume of 100 μL of the peptide solutions concentrated at 2048 μg/mL in PBS was then added to the first wells of the plates, and serial dilutions were performed by transferring 50 μL from one well to the next up to the twelfth well of the plates in order to obtain final peptide concentrations between 1024 μg/mL and 0.5 μg/mL. Then, 50 μL of the washed erythrocyte suspension was deposited in the wells. The final volumes were 100 μL in the wells. The microplates were then incubated at 37 °C for 1 h and centrifuged at 2500× *g* for 10 min.

After incubation, the supernatants from the plates (50 μL) were then transferred into Falcon^®^ polystyrene clear flat bottom 96-well microplates (Corning Inc., Corning, NY, USA), and hemoglobin leakage from erythrocytes was quantified by measuring their absorbance at 405 nm with a BioTek Synergy H1 spectrophotometer (Agilent, Santa Clara, CA, USA). TritonX-100 solution (Sigma-Aldrich, St. Louis, MI, USA) (1%) and PBS were used as positive and negative controls, respectively. PBS was also used as a blank. Three independent replications, each including three technical replicates, were performed. The percentage of hemolysis was calculated with the following equation:$${\%}_{hemolysis}=\frac{{(A}_{sample}-A_{PBS})}{{(A}_{TritonX\text{-}100}-A_{PBS})}\times 100$$

### 4.4. In Vitro Digestibility Assays in a Dynamic Gastro-Intestinal Tract Model

The dynamic GI TIM-1 Model (TNO Nutrition and Food Research Institute, Zeist, The Netherlands), as described by Minekus et al. [69], was used to evaluate the digestibility of peptide analogs 6 and 8. The pH in the stomach began at 4.5 and decreased gradually over time as 0.5 mol/L of HCl was injected. After 20 min of digestion, the pH in the stomach dropped to 3.6, and after 40 min it dropped further to 2.7. The pH continued to decrease, reaching 2.0 after 60 min and finally reaching 1.7 after 120 min. The pH of the duodenal, jejunal, and ileal compartments was adjusted to 6, 6.8, and 7.2, respectively, by injecting 0.5 mol/L sodium bicarbonate solution. Gastric secretions consisted of pepsin (0.163 mg/mL) from porcine gastric mucosa (EC 3.4.23.1; Sigma-Aldrich Canada Ltd., Oakville, ON, Canada) and lipase (0.133 mg/mL) from *Rhizopus oryzae* (EC 3.1.1.3; Amano Pharmaceuticals, Nagoya, Japan), and both of them were added to an electrolyte solution (NaCl, 6.2 g/L; KCl, 2.2 g/L; CaCl_2_, 0.3 g/L; NaHCO_3_, 1.5 g/L), which was delivered at a flow rate of 0.25 mL/min. 4% pancreatin solution (8xUSP, Pancrex V powder; Paines and Byrne, Greenford, UK) and 4% porcine bile extract (Sigma-Aldrich Canada Ltd.), as well as the small intestine electrolyte solution (NaCl, 5.0 g/L; KCl, 0.6 g/L; CaCl_2_, 0.3 g/L; pH 7.0), consisted of duodenal secretions and were injected at 0.25, 0.5, and 0.25 mL/min, respectively.

Before starting the digestion process, 1 mL of trypsin solution (2 mg/mL) was added to the duodenal compartment, and 25 mg of synthetic peptide was dissolved in 310 mL of distilled water. A volume of 10 mL of this peptide solution (the initial meal) was withdrawn, and the remaining 300 mL were injected into the stomach compartment. Samples (10 mL) were withdrawn from the stomach, duodenum, and ileal-delivered effluent at intervals of 0, 20, 40, 60, and 120 min during the digestion process. They were then heated at 70 °C for 15 min in a water bath to stop enzymatic activity, and then kept frozen at −18 °C until use. Two independent digestions for peptide analogs 6 and 8 and two independent blank digestions (meals consisting of distilled water only) were performed.

### 4.5. Analysis of GI TIM-1 Fractions

#### 4.5.1. Fast Protein Liquid Chromatography (FPLC)

To determine the digestibility of peptides 6 and 8 in the human digestive tract, the 10 mL samples from the TIM-1 compartments withdrawn at different sampling times, including the initial meal, were lyophilized, resolubilized in 2 mL of sodium phosphate buffer (0.05 M) (pH 7.2), and filtered over 0.45 uM. To detect the peptides in these samples, a HiTrap SP FF cation exchange column (1 mL, GE, Mississauga, ON, Canada) was used on a FPLC system (Akta Avant, GE Healthcare, Baie-D’Urfé, QC, Canada), and 500 uL injections were performed for each sample. The equilibration buffer consisted of sodium phosphate buffer (0.05 M) (pH 7.2). Before each injection, the column was equilibrated with six column volumes of equilibration buffer.

For each run, after the sample injection, the column was washed with 5 column volumes of equilibration buffer, and a linear NaCl gradient was then used until a NaCl concentration of 0.5 M was achieved over 6 column volumes. A second linear gradient was then applied to achieve a 0.9 M NaCl concentration over two column volumes. The column was then washed with equilibration buffer containing 1 M NaCl for 5 column volumes, and finally, it was equilibrated with 6 column volumes of equilibration buffer. The absorbance was recorded at 214 nm.

#### 4.5.2. Liquid Chromatography with Tandem Mass Spectrometry (LC-MS/MS)

Mass spectrometry analyses were performed by the Proteomics Platform of the CHU de Québec Research Center (Québec, QC, Canada). An amount of 100 μL of peptide samples from the TIM-1 comparments were desalted on a C18 Empore filter (Stage-Tip), and 205 nm absorbance measurements (Nanodrop, Thermo Fischer, Saint-Laurent, QC, Canada) were used to adjust the concentration for the analysis of 1 μg of material by LC-MS/MS. The experiments were performed with a Dionex UltiMate 3000 nanoRSLC chromatography system (Thermo Fisher Scientific, San Jose, CA, USA) connected to an Orbitrap Fusion mass spectrometer (Thermo Fisher Scientific, San Jose, CA, USA) equipped with a nanoelectrospray ion source. Peptides were trapped at 20 μL/min in loading solvent (2% acetonitrile (ACN), 0.05% trifluoroacetic acid (TFA)) on a 5 mm × 300 μm C18 pepmap cartridge (Thermo Fisher Scientific, San Jose, CA, USA) for 5 min. Then, the pre-column was switched online with a 50 cm × 75 µm internal diameter separation column (Pepmap Acclaim column, Thermo Fisher Scientific, San Jose, USA), and the peptides were eluted with a linear gradient from 5–40% solvent B (A: 0.1% TFA, B: 80% ACN, 0.1% TFA) in 30 min at 300 nL/min (60 min total runtime).

Mass spectra were acquired using a data-dependent acquisition mode using Thermo XCalibur software version 4.1.50. Full-scan mass spectra (350 to 1800 *m/z*) were acquired in the orbitrap using an AGC target of 4e5, a maximum injection time of 50 ms, and a resolution of 120,000. Internal calibration using lock mass on the *m/z* 445.12003 siloxane ion was used. Each MS scan was followed by the acquisition of fragmentation MSMS spectra of the most intense ions for a total cycle time of 3 s (top speed mode). The selected ions were isolated using the quadrupole analyzer with 1.6 *m/z* windows and fragmented by higher-energy collision-induced dissociation (HCD) with 35% collision energy. The resulting fragments were detected by the linear ion trap at a rapid scan rate with an AGC target of 1e4 and a maximum injection time of 50 ms. Dynamic exclusion of previously fragmented peptides was set for a period of 30 s and a tolerance of 10 ppm.

MGF peak list files were created using Proteome Discoverer Software (version 2.3, Thermo). MGF files were then analyzed using Mascot (Matrix Science, London, UK; version 2.5.1). Mascot was set up to search a contaminant database and the peptide sequences, assuming a non-specific digestion using “none” as the enzyme parameter. Mascot was searched with a fragment ion mass tolerance of 0.60 Da and a parent ion tolerance of 10.0 PPM. The carbamidomethyl of cysteine was specified in Mascot as a fixed modification. Deamidation of asparagine and glutamine and oxidation of methionine were specified in Mascot as variable modifications. Two missed cleavages were allowed.

### 4.6. Statistical Analysis

All statistical work was done with the SAS^®^ Studio software (version 3.8, SAS Institute Inc., Cary, NC, USA). One-way ANOVA and Tukey’s HSD test were used to identify significant differences between treatments. Results were considered statistically significant when *p* < 0.05.

## 5. Conclusions

In conclusion, the results presented in this study show that the antimicrobial activity of SJGAP and analogs thereof is negatively affected by the presence of cations, especially divalent cations, which totally suppress it. Furthermore, the peptides were found to be much more active at acidic pH, showing MICs as low as 2 ug/mL in the case of analogs 5 and 8 at pH 5, even though their bactericidal power is decreased at these low pH values. Therefore, it seems that further mechanistic studies are needed to explain this decrease in bactericidal activity despite the much higher inhibitory activity of these peptides in acidic conditions. This study also showed that these peptides did not have significant hemolytic activity when used at their MICs (and even higher concentrations) and that analogs 6 and 8 were very rapidly degraded in a dynamic model of the human gastrointestinal tract, both of which results point to their safety. Thus, from a clinical point of view, the potential of these peptides is encouraging, and it would be relevant to continue the research by testing them on strains of clinical interest. Moreover, the results obtained in this study indicate a very interesting potential for use in the food sector. The fast degradation in the gastrointestinal tract prevents the accumulation of residues in the body and makes their negative impact on the intestinal microbiota very unlikely, making them promising candidates as food preservatives. Efficacy tests in complex food matrices could certainly further document their potential for use in the food industry.

## Figures and Tables

**Figure 1 antibiotics-12-01410-f001:**
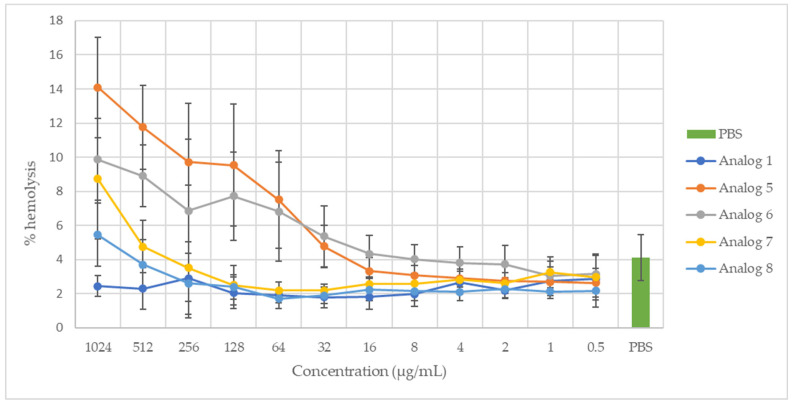
Hemolytic activity of peptide analogs 1, 5, 6, 7, and 8 at concentrations ranging from 1024 μg/mL to 0.5 μg/mL, with PBS used as a negative control and TritonX-100 (1%) as a positive control (100% hemolysis).

**Figure 2 antibiotics-12-01410-f002:**
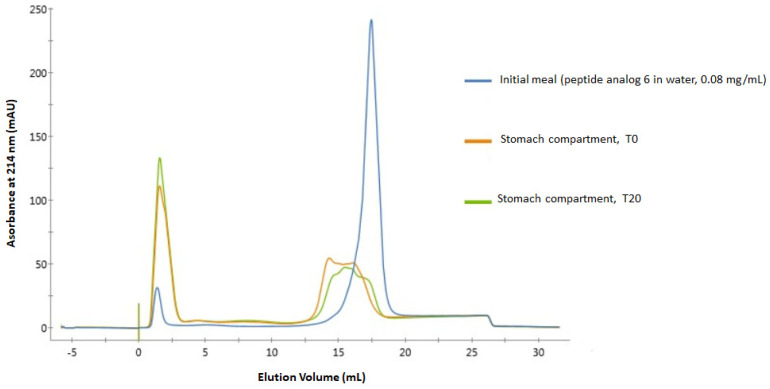
FPLC spectra of peptide analog 6 samples were withdrawn from the initial meal (blue), the stomach compartment at T0 (orange), and the stomach compartment after 20 min (green).

**Figure 3 antibiotics-12-01410-f003:**
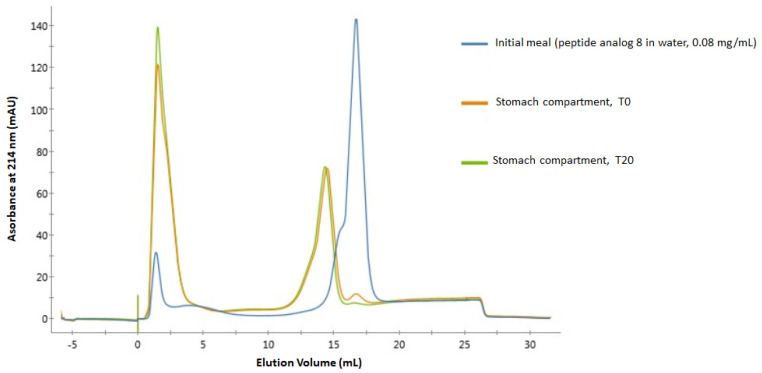
FPLC spectra of peptide analog 8 samples were withdrawn from the initial meal (blue), stomach compartment at T0 (orange), and stomach compartment after 20 min (green).

**Table 1 antibiotics-12-01410-t001:** Minimal inhibitory concentrations (MICs) and minimal bactericidal concentrations (MBCs) of peptide analogs against *E. coli* ATCC 11229 in the presence of 1, 5, and 10 mM of CaCl_2_ and MgCl_2_.

Antimicrobial Compounds	Antibacterial Activity (MIC; MBC) (μg/mL)						
	CaCl_2_			MgCl_2_			MHB (Ctrl −)
	1 mM	5 mM	10 mM	1 mM	5 mM	10 mM	
Analog 1	n.a. ^2^	n.a.	n.a.	n.a.	n.a.	n.a.	n.a.
Analog 5	n.a.	n.a.	n.a.	n.a.	n.a.	n.a.	16; 32
Analog 6	n.a.	n.a.	n.a.	n.a.	n.a.	n.a.	16; 32
Analog 7	n.a.	n.a.	n.a.	n.a.	n.a.	n.a.	n.a.
Analog 8	n.a.	n.a.	n.a.	n.a.	n.a.	n.a.	32; 64
Chloramphenicol ^1^	8	8	8	8	8	8	8

^1^ Only MIC values are shown for chloramphenicol. ^2^ No activity.

**Table 2 antibiotics-12-01410-t002:** Minimal inhibitory concentrations (MICs) and minimal bactericidal concentrations (MBCs) of peptide analogs against *E. coli* ATCC 11229 in the presence of 50, 150, and 300 mM of NaCl and KCl.

Antimicrobial Compounds	Antibacterial Activity (MIC; MBC) (μg/mL)						
	NaCl			KCl			MHB (Ctrl −)
	50 mM	150 mM	300 mM	50 mM	150 mM	300 mM	
Analog 1	n.a. ^2^	128; n.b.a. ^3^	128; n.b.a.	n.a.	n.a.	n.a.	n.a.
Analog 5	32; 32	n.a.	n.a.	32; 32	n.a.	n.a.	16; 32
Analog 6	64; 64	n.a.	n.a.	32; 64	n.a.	n.a.	16; 32
Analog 7	64 *; n.b.a.	128 *; n.b.a.	n.a.	64 *; n.b.a.	128 *; n.b.a.	n.a.	n.a.
Analog 8	n.a.	n.a.	n.a.	n.a.	n.a.	n.a.	32; 64
Chloramphenicol ^1^	8	4	4	8	4	2	8

^1^ Only MIC values are shown for chloramphenicol. ^2^ No activity; ^3^ no bactericidal activity. * A very weak bacterial growth could be detected. See main text for details.

**Table 3 antibiotics-12-01410-t003:** Minimal inhibitory concentrations (MICs) and minimal bactericidal concentrations (MBCs) of peptide analogs against E. coli ATCC 11229 at different pH values and after a heating treatment (100 °C, 15 min).

Antimicrobial Compounds	Antibacterial Activity (MIC; MBC) (μg/mL)					
	pH 5	pH 6	pH 7	pH 8	100 °C, 15 min	MHB (Ctrl −)
Analog 1	4; n.b.a.	16; n.b.a.	64; 128	128; 128	n.a. ^2^	n.a.
Analog 5	2; 8	8; 16	16; 128	32; 128	n.a.	16; 32
Analog 6	8; n.b.a.	16; 128	16; 32	16; 16	32; 64	16; 32
Analog 7	8 *; 8	8 *; 32	n.a.	n.a.	n.a.	n.a.
Analog 8	2; n.b.a. ^3^	16; 32	64; 64	128; n.b.a.	64; 128	32; 64
Chloramphenicol ^1^	0.5	2	4	4	8	8

^1^ Only MIC values are shown for chloramphenicol. ^2^ No activity; ^3^ no bactericidal activity. * A very weak bacterial growth could be detected. See main text for details.

**Table 4 antibiotics-12-01410-t004:** Sequences, net charges, pIs, molar weights, and GRAVY indexes of the peptide analogs used in this study.

Analog Identification	Sequence	Net Charge	pI	Molar Weight (g/mol)	GRAVY Index
(1) SJGAP	VKVGINGFGRIGRLVTRAAFHGKKVEIVAIND	+4	11.4	3436.07	0.272
(5)	VKVGINGFGRIGRLVTRAAFHGKKVAIVAINA	+6	12.4	3334.02	0.603
(6)	VKVGINGFGRIGRLVTRAAFHGKKVKIVAINK	+8	12.5	3448.21	0.247
(7)	VKVGINGFGRIGRLVTRLLFHGKKVEIVLIND	+4	11.4	3562.21	0.459
(8)	VKVGINGFGRIGRLVTRAAFHGKKVEIVAIND-NH_2_	+5	11.9	3435.08	0.272

## Data Availability

The data presented in this study are available on request from the corresponding author.

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
