# Peer review of "Assessing the Activity under Different Physico-Chemical Conditions, Digestibility, and Innocuity of a GAPDH-Related Fish Antimicrobial Peptide and Analogs Thereof"

_antibiotics, 2023, doi:10.3390/antibiotics12091410_

Round 1
Reviewer 1 Report
Comments: The article entitled “Impact of pH and Cations on the Antimicrobial Activity of a GAPDH-related Fish Peptide and Analogs Thereof, Determination of their Hemolytic Activity and Study of their In Vitro Digestibility” is overall a good scientific effort that has been made by Lucie Beaulieu et al. The background and purpose of this study is interesting, also, the article has been properly handled. I believe that this article will fulfill the quality to be published in Journal of Antibiotics if the authors include all the suggested modifications.
Recommendation: Accept with minor revision
1. The title is too long, modification is required
2. The last paragraph of introduction “The aim of this study is to further document the potential application of SJGAP and chemical analogs thereof as therapeutic molecules or food antimicrobial agents. To do this, several objectives were pursued. First, (i) the impact of different physicochemical conditions (presence of monovalent and divalent cations, pH and heating treatment) on the antimicrobial activity of SJGAP and its analogs was determined, and (ii) their toxicity was assessed by measuring their hemolytic activity. This allowed to (iii) establish relationships between the chemical structure of these analogs and their stability under different physicochemical conditions. Finally, (iv) the susceptibility of the peptides to digestive enzymes was evaluated in a dynamic model of the human gastro-intestinal tract, providing information on their safety in relation to their possible accumulation and potential routes of administration.” need to be removed and must be present in a very concise form in the abstract.
3. The authors reported “SJGAP [21] was selected as the reference AMP for this study. Native SJGAP and four peptide analogs, which were designed as previously described [22], were synthesized by the external company Bio Basic (Markham, Canada) with a > 95% purity, as confirmed by HPLC/MS reports.” So the authors need to provide synthetic evidence with spectroscopic data in the article of these claimed compounds which are missing. Also, the HPLC/MS data need to be incorporated in the manuscript.
4. The biological activates of these compounds with 95% purity need to be justified as the 5% impurity must have a positive or negative effect over the biological activities. How we exclude the role of 5% impurity?
5. As these are claimed to be synthetic compounds so all the spectroscopic data must be incorporated. So, it is advised to include all these supporting synthetic data in the supplementary file.
6. These peptides can be future therapeutic agents; however, few queries need to be addressed. Such as
7. Why the authors have not targeted the peptides from medicinal plants sources?
8. The major disadvantage of the Long AMP is the sequences that are economically and synthetically challenging for pharmaceutical scale synthesis, how they will respond to this issue.
9. Another disadvantage of the peptides is “AMP sequences often contain complex disulfide bridges and CCKs which are challenging to be synthetically mimicked”.
10. What could be the possible binding site of the peptide on bacteria surface?
11. Check the whole article for typographical mistakes.
12. Some old but basic models can also be added like the disc diffusion method, if needed
13. The proteolytic instability is also a major issue with peptides, if possible the significant concentration of peptides can be tested in rodent model like mice or rat and bioavailability need to be traced in a biological environment.
Require minor editing of English language
Author Response
Reviewer # 1:
1. New title proposal: Assessing the activity under different physico-chemical conditions, digestibility and innocuity of a GAPDH-related Fish Antimicrobial Peptide and Analogs Thereof.
2. We thought this paragraph clearly indicated the content of the article to make it easier to read. We used this formula in our last article published in Antibiotics [1].
3. HPLC/MS reports of the synthetic peptides provided by the external company Bio Basic can be included in the supplementary information section.
4. Synthetic peptides with a >95% purity are regularly used in the literature for this type of study [1-5]. At the low peptide concentrations used, 5% (or less) impurities have a very negligible impact on antimicrobial activity and ultra-pure peptides (>98%) are generally used for in vivo and crystallography studies.
5. Please see point 3.
7. Several research groups are interested in antimicrobial peptides from plants. Our research group specializes in bioactive molecules of marine origin, which explains the choice of peptides selected for this study. This study also follows on from our last published work on the same peptides [1].
8. Indeed, the cost of synthesizing peptides is one of the challenges associated with their large-scale use. However, advances in research and technology have already greatly reduced the cost of peptide synthesis [6] and many synthetic antimicrobial peptides are currently tested in clinical studies [7]. Moreover, once the sequence of an antimicrobial peptide of interest is known, biosynthesis can be carried out using microbial expression systems.
9. The peptide sequences used do not contain cysteine residues, thus limiting the presence of disulfide bridges.
10. Another manuscript is currently being written to study in greater detail the interactions between these peptides and microbial membranes. It is relevant to note, however, that antimicrobial peptides often do not have specific molecular targets, but interact with phospholipids and other components of microbial membranes, causing its permeabilization (please see lines 280-285).
13. Sensitivity to proteolytic enzymes may be highly desirable for some applications (in the food industry, for example). We have addressed this point in the discussion. Indeed, the fact that they are rapidly digested in the gastrointestinal tract limits the possibility of accumulation and is an important point indicating peptide safety [8, 9].
Thank you very much for your comments and suggestions !
References
- Cashman-Kadri, S., et al., Determination of the Relationships between the Chemical Structure and Antimicrobial Activity of a GAPDH-Related Fish Antimicrobial Peptide and Analogs Thereof. Antibiotics, 2022. 11(3): p. 297 DOI: 10.3390/antibiotics11030297.
- Wang, Q., et al., HJH-1, a Broad-Spectrum Antimicrobial Activity and Low Cytotoxicity Antimicrobial Peptide. Molecules (Basel, Switzerland), 2018. 23(8) DOI: 10.3390/molecules23082026.
- Chu, H.-L., et al., Antimicrobial Peptides with Enhanced Salt Resistance and Antiendotoxin Properties. International journal of molecular sciences, 2020. 21(18): p. 6810 DOI: 10.3390/ijms21186810.
- Dong, N., et al., Short Symmetric-End Antimicrobial Peptides Centered on β-Turn Amino Acids Unit Improve Selectivity and Stability. Frontiers in Microbiology, 2018. 9(2832) DOI: 10.3389/fmicb.2018.02832.
- Vishweshwaraiah, Y.L., et al., Rational design of hyperstable antibacterial peptides for food preservation. npj Science of Food, 2021. 5(1) DOI: 10.1038/s41538-021-00109-z.
- Lima, P.G., et al., Synthetic antimicrobial peptides: Characteristics, design, and potential as alternative molecules to overcome microbial resistance. Life Sci, 2021. 278: p. 119647 DOI: 10.1016/j.lfs.2021.119647.
- Dijksteel, G.S., et al., Review: Lessons Learned From Clinical Trials Using Antimicrobial Peptides (AMPs). Frontiers in microbiology, 2021. 12: p. 616979 DOI: 10.3389/fmicb.2021.616979.
- Naimi, S., et al., Fate and Biological Activity of the Antimicrobial Lasso Peptide Microcin J25 Under Gastrointestinal Tract Conditions. Front Microbiol, 2018. 9: p. 1764 DOI: 10.3389/fmicb.2018.01764.
- Ben said, L., et al., Antimicrobial Peptides: The New Generation of Food Additives, in Encyclopedia of Food Chemistry, L. Melton, F. Shahidi, and P. Varelis, Editors. 2019, Academic Press: Oxford. p. 576-582.
Reviewer 2 Report
This is an interesting paper in an important area.
I appreciate that the authors have clearly included the number of independent and technical replicates.
There are a few places that a decimal place should be used instead of a comma for this paper. Such as in the bottom left of table 3 (check throughout).
In table 1, it would be helpful to put n.a. in every spot needed. Initially, I thought that the no activity was only for analog 6 at 5 mM… This will help with clarity of the table if I am correct that it was tested for each analog at each concentration of CaCl2 and MgCl2.
In Figure 1, it would be helpful to label the legend using the term analog to be consistent with the tables.
Why were only analogs 6 and 8 tested for the gastro-intestinal model?
I appreciate how the stats were done.
I like how the relevance of the findings are clear.
Author Response
Reviewer # 2:
- Decimal points have been included in tables 3 and 4 instead of decimal commas.
- Table 1 has been modified as requested.
- Figure 1 has been modified as requested.
- Only analogs 6 and 8 were tested in the gastro-intestinal model, because analog 6 showed to be the most active peptide while showing no significant hemolytic activity, and analog 8 is the only one bearing a C-terminal amidation, which is known to enhance proteolytic stability in some cases [1, 2]. Please see lines 452-457.
We also checked the sensitivity of the peptides to digestive enzymes using in silico prediction methods, and all the peptides had similar cleaving sites for the enzymes used in the experiments.
Thank you very much for your comments and suggestions !
References
- Heavner, G.A., et al., Biologically active analogs of thymopentin with enhanced enzymatic stability. Peptides, 1986. 7(6): p. 1015-1019 DOI: https://doi.org/10.1016/0196-9781(86)90131-2.
- Brinckerhoff, L.H., et al., Terminal modifications inhibit proteolytic degradation of an immunogenic mart-127–35 peptide: Implications for peptide vaccines. International Journal of Cancer, 1999. 83(3): p. 326-334 DOI: https://doi.org/10.1002/(SICI)1097-0215(19991029)83:3<326::AID-IJC7>3.0.CO;2-X.
Reviewer 3 Report
The study by Cashman-kadri reported the optimal conditions for the activity of a fish AMP, mainly focusing on the impact of pH and cations. It is known that conditions, either in vivo or in vitro, are essential for the functions of many AMPs and other “proteins”. The reviewer agrees that figuring out the factors affecting AMPs' activity is important for their future applications. Following comments are listed for authors to improve their manuscript.
1) The title is too long and not appropriate.
2) The only bacteria used are E. coli. More species with both Gram+/- are required.
3) The concentrations for CaCl2, MgCl2, NaCl, and KCl should be discussed in details and explain why those concentrations are used. Moreover, conditions with both cations presented, such as combinations with CaCl2 and MgCl2, are also needed since those cations also effects on the membrane status of bacteria.
4) Another point is the osmolality. The authors should calculate or directly measure the Osmo in order to clarify if this factor is also related to the activity of the AMP used.
5) Discussion is a little loose. The key sentence should be listed to show a simple logic.
Need further careful proofreading.
Author Response
Reviewer # 3:
- New title proposal:
Assessing the activity under different physico-chemical conditions, digestibility and innocuity of a GAPDH-related Fish Antimicrobial Peptide and Analogs Thereof
- In an article published last year in Antibiotics [1], we tested the activity of all these peptide analogs on several Gram + and Gram - bacteria, as well as on several fungal strains. In this article, we performed the tests on E. coli ATCC 11229, as this is the strain that has proved most sensitive to peptides in our previous work. As the aim was to study the stability of the peptides, and not to precisely quantify their activity against different microorganisms (which has already been done), we decided to use the most relevant bacterial strain to facilitate comparisons of antimicrobial activity under the different conditions tested. Please see lines 524-526.
- Information regarding salt concentrations have been added in the discussion (please see lines 313-316).
- We understand that osmolality calculations may be relevant to the study of peptide mechanisms of action under certain specific conditions. However, in our study, we relied on the protocols of numerous published scientific articles to assess the impact of the presence of cations on peptide activity [2-5]. In these numerous articles, no consideration was given to the osmolality of the media tested. Moreover, since the same base medium (MHB) was used in all the tests, and the concentrations of the salts used are comparable, the results obtained highlight differences in activity according to the cations present, rather than differences in osmolality.
Thank you very much for your comments and corrections !
References
- Cashman-Kadri, S., et al., Determination of the Relationships between the Chemical Structure and Antimicrobial Activity of a GAPDH-Related Fish Antimicrobial Peptide and Analogs Thereof. Antibiotics, 2022. 11(3): p. 297 DOI: 10.3390/antibiotics11030297.
- Chu, H.-L., et al., Antimicrobial Peptides with Enhanced Salt Resistance and Antiendotoxin Properties. International journal of molecular sciences, 2020. 21(18): p. 6810 DOI: 10.3390/ijms21186810.
- Wu, G., et al., Effects of cations and pH on antimicrobial activity of thanatin and s-thanatin against Escherichia coli ATCC25922 and B. subtilis ATCC 21332. Current microbiology, 2008. 57(6): p. 552-7 DOI: 10.1007/s00284-008-9241-6.
- Dong, N., et al., Short Symmetric-End Antimicrobial Peptides Centered on β-Turn Amino Acids Unit Improve Selectivity and Stability. Frontiers in Microbiology, 2018. 9(2832) DOI: 10.3389/fmicb.2018.02832.
- Kwon, J.Y., et al., Mechanism of action of antimicrobial peptide P5 truncations against Pseudomonas aeruginosa and Staphylococcus aureus. AMB Express, 2019. 9(1): p. 122 DOI: 10.1186/s13568-019-0843-0.
Reviewer 4 Report
To broaden the search for this manuscript, I suggest that the keywords be rewritten, since many of these words are already in the title;
The introduction is well written, however some paragraphs are too long, I suggest they be split;
In line 53 it is mentioned that fish are very good sources of natural MPAs. What species of fish? Or are all species used as sources?
At the beginning of the discussion thread is looking like an introduction. I suggest improvements in the writing of this primary topic in the manuscript;
In the materials and methods there are long paragraphs, I suggest improvements
Author Response
Reviewer # 4:
- New title proposal:
Assessing the activity under different physico-chemical conditions, digestibility and innocuity of a GAPDH-related Fish Antimicrobial Peptide and Analogs Thereof
Keywords have been added.
- Longer paragraphs in the introduction have been split.
- This is a general remark based on the scientific literature on marine antimicrobial peptides. Indeed, many published articles concern the identification and characterization of antimicrobial peptides from different fish species. There are even specific literature reviews on the subject [1-4].
- We agree that the first paragraph of the discussion is general and introductory. We felt that this allowed us to present some more general information supporting the relevance of the specific points presented in the discussion. We deleted last sentence of the first paragraph of the discussion, which does not really support results presented and discussed in the paper.
- Some sections of the Materials & Methods have been modified to shorten some of the longer paragraphs (sections 4.3, 4.4, 4.5.1, 4.5.2).
Thank you very much for your comments and corrections !
References
- Masso-Silva, J.A. and G. Diamond, Antimicrobial peptides from fish. Pharmaceuticals (Basel), 2014. 7(3): p. 265-310 DOI: 10.3390/ph7030265.
- Rajanbabu, V. and J.Y. Chen, Applications of antimicrobial peptides from fish and perspectives for the future. Peptides, 2011. 32(2): p. 415-20 DOI: 10.1016/j.peptides.2010.11.005.
- Shabir, U., et al., Fish antimicrobial peptides (AMP's) as essential and promising molecular therapeutic agents: A review. Microb Pathog, 2018. 114: p. 50-56 DOI: 10.1016/j.micpath.2017.11.039.
- Najafian, L. and A.S. Babji, A review of fish-derived antioxidant and antimicrobial peptides: their production, assessment, and applications. Peptides, 2012. 33(1): p. 178-85 DOI: 10.1016/j.peptides.2011.11.013.
Round 2
Reviewer 3 Report
This reviewer has no further comments on this revised manuscript.
Reviewer 4 Report
Suggestions have been made, and I am in favor of publishing the manuscript.